# Methods for Assessment and Monitoring of Light Pollution around Ecologically Sensitive Sites

**DOI:** 10.3390/jimaging5050054

**Published:** 2019-05-18

**Authors:** John C. Barentine

**Affiliations:** 1International Dark-Sky Association, 3223 N. First Avenue, Tucson, AZ 85719, USA; john@darksky.org; Tel.: +1-520-347-6363; 2Consortium for Dark Sky Studies, University of Utah, 375 S 1530 E, RM 235 ARCH, Salt Lake City, UT 84112-0730, USA

**Keywords:** conservation, light pollution, artificial light at night, imaging, radiometry, skyglow, darkness, night

## Abstract

**Simple Summary:**

Imaging and photometric techniques are used to characterize the brightness of nighttime conditions in protected areas in support of conservation efforts.

**Abstract:**

Since the introduction of electric lighting over a century ago, and particularly in the decades following the Second World War, indications of artificial light on the nighttime Earth as seen from Earth orbit have increased at a rate exceeding that of world population growth during the same period. Modification of the natural photic environment at night is a clear and imminent consequence of the proliferation of anthropogenic light at night into outdoor spaces, and with this unprecedented change comes a host of known and suspected ecological consequences. In the past two decades, the conservation community has gradually come to view light pollution as a threat requiring the development of best management practices. Establishing those practices demands a means of quantifying the problem, identifying polluting sources, and monitoring the evolution of their impacts through time. The proliferation of solid-state lighting and the changes to source spectral power distribution it has brought relative to legacy lighting technologies add the complication of color to the overall situation. In this paper, I describe the challenge of quantifying light pollution threats to ecologically-sensitive sites in the context of efforts to conserve natural nighttime darkness, assess the current state of the art in detection and imaging technology as applied to this realm, review some recent innovations, and consider future prospects for imaging approaches to provide substantial support for darkness conservation initiatives around the world.

## 1. Introduction

Over a century after the introduction of electric lighting, the use of artificial light at night (ALAN) has become a truly global phenomenon. Remote sensing observations of the Earth at night indicate that a majority of humans now live in places where the night sky at the zenith is measurably affected by light pollution [1]. Indications of light as seen from space have grown in recent years at a global annual rate of ~2% per year in terms of both lit area and total radiance, with large departures from the global average seen for many countries [2]. A subset of people never experience conditions like a true night, as the luminosity of the night sky never drops below the situation represented by the luminance of a pristine sky at the end of evening nautical twilight (~1.4 mcd m${}^{-2}$). Average increases in the consumption of ALAN closely mirror the global average rate of gross domestic product (GDP) growth, indicating an economic rebound effect in the outdoor lighting sector. This appears to be enabled by the rapid proliferation of energy-efficient solid-state lighting technologies in the past decade, which have lowered the cost of providing outdoor lighting and fueled consumer demand.

While the use of ALAN clearly conveys certain social and economic benefits to humans [3], it is associated with a number of negative environmental externalities. An extensive body of evidence documents a host of known and suspected biological and ecological hazards associated with the use of ALAN. Effects due to ALAN exposure have been observed among birds [4,5,6,7], fishes [8,9,10], mammals [11,12,13], reptiles [14,15,16], invertebrates [17,18,19,20,21], and plants [22,23]. It is now clear that ALAN has the potential to disrupt biological processes relying on the light cues of daily and seasonal rhythms, including foraging behaviors [24,25,26,27], the timing of emergence [28,29,30,31], reproduction [32,33,34], and communication [35,36,37]. There are wide-ranging implications for the ecological harms caused by ALAN, such as alteration of predator–prey interactions and diminishing the resiliency of food webs [38,39,40], disruption of ecosystem services [41,42,43], and threats to biodiversity [44,45].

Parks, nature reserves, and similar protected areas around the world now find themselves at the forefront of the effort to preserve what remains of the planet’s natural nighttime darkness. In many instances, they are equally motivated by the economic development potential of sustainable forms of tourism. These tourism modalities include “astrotourism” catering to visitors who wish to view dark night skies [46,47], as well as wildlife ecotourism, in which visitors seek to view nocturnal species in natural nighttime conditions. However, best practices for managing this resource are still nascent, and the literature on the subject is scant and tends to focus on tourism rather than systematic conservation practices [48]. Where wildlife is concerned, there is some evidence that lighting provided for the convenience of ecotourists is potentially detrimental to the species they come to view [49]; however, other work suggests that actively managing nighttime visitor interactions with wildlife can minimize potential impacts while increasing visitor satisfaction [50].

One main conservation method establishes and promotes the status of certain protected landscapes on the basis of the quality of their dark night skies; notable examples include the International Dark-Sky Association (IDA) International Dark Sky Places Program [51], the Royal Astronomical Society of Canada (RASC) Dark Sky Site Designations [52], and the Starlight Foundation certification program [53]. Each program establishes its own means of characterizing the darkness of designation candidates through a series of criteria involving both objective measurements and subjective impressions of the quality of nighttime darkness at the site. There is a clear need to both assess the current state of darkness in these places and to monitor its evolution in time [54]. The same is true of cities seeking accreditation as IDA International Dark Sky Communities: it is important to know the magnitude of light pollution in and near cities to assess the efficacy of public policies and lighting interventions intended to reduce cities’ impacts not only on urban areas and places lacking environmental protections, but also on adjacent protected areas. Sensing of light in the nocturnal environment helps to determine the conservation state of these places, identify emerging threats, and suggest land management actions to preserve natural nighttime darkness.

This review is organized as follows. First, existing approaches to characterizing the nighttime photic environment in ecologically sensitive areas are reviewed in Section 2. Next, Section 3 highlights some recent innovations in this arena. Lastly, potential future development of new approaches that may enable land managers to address light pollution threats more proactively is considered in Section 4, along with some concluding remarks.

## 2. Current Methods

Site characterization generally takes a two-pronged approach, which involves looking both down at the Earth with orbital remote sensing and up at the night sky from the Earth’s surface with various radiometric sensors. While the primary interest in describing and ranking places according to the quality of the nocturnal environment is served by measuring the brightness of the night sky, a perspective that is only ground-based overlooks crucial information. This is particularly true in consideration of the ecological consequences of ALAN use [55].

A comprehensive review of various approaches to measuring the brightness of the night sky was recently published in [56]. Additionally, several methods should be noted for qualitatively gauging the brightness of the night sky, such as estimates of naked-eye limiting visual magnitude [57] and the Bortle sky quality scale [58]. Although these estimates are subjective and observer-specific, there is some evidence that they map to objective measures in a repeatable way [59]. They also provide a valuable source of both public engagement in night conservation issues, as well as opportunities for citizen science participation.

### 2.1. Remote Sensing of Upward Radiance

A first-order guess about the anthropogenic contribution to the brightness of the night sky at any given point on Earth begins with a determination of the sources of light on the ground; when paired with a radiative transfer model describing how that light propagates through the Earth’s atmosphere [60,61,62,63,64,65,66,67], one can infer the amount of scattered light seen toward the zenith.

Some fraction of light from ground sources completely escapes the atmosphere and can be detected from Earth orbit. The first remote sensing detections of Earth’s “night lights” were made in the 1960s, and the detection capacity increased significantly in the following decade through the deployment of satellites in the U.S. Defense Meteorological Satellite Program (DMSP). Initially recording night lights on photographic film, the system began returning digital data in 1991–1992. The DMSP Operational Linescan System (OLS) is a scanning radiometer that uses a photomultiplier giving sensitivity over the wavelength range of 0.44–0.94 μm to a spectral radiance limit of 1 nW cm${}^{-2}$ sr${}^{-1}$μm${}^{-1}$. OLS data have been used to generate global night light maps since the mid-1990s [68,69]. Although there are known calibration issues with OLS, including detector saturation by bright cities [70,71], studies of OLS data have yielded useful results indicating patterns of land use and human activities on the nighttime Earth, as well as the first calibrated, global anthropogenic radiance map [72]. The data have also been adapted for use in estimating the brightness of the night sky from various locations for which ground-based measurements are not otherwise available [73,74,75].

While DMSP revolutionized night lights studies and offered the first truly worldwide view of the problem of light pollution, its calibration issues and relatively poor spatial resolution (~3 km pixel${}^{-1}$ at the Earth’s surface in “smooth” mode) ultimately limit the utility of OLS data. A new generation of Earth observatory platforms has improved the situation. In 2011, the U.S. National Aeronautics and Space Administration (NASA) launched the Suomi National Polar-orbiting Partnership satellite. Among the Suomi instruments is the Visible Infrared Imaging Radiometer Suite (VIIRS), a “whisk broom” scanning radiometer [76,77]. Of the VIIRS passbands, only one achieves sufficient sensitivity to make meaningful quantitative measurements of night lights: the Day-Night Band (DNB), which is sensitive to light between 0.5 and 0.9 μm and has a ground spatial resolution of ~750 m pixel${}^{-1}$ and minimum detectable radiance of 3 nW cm${}^{-2}$ sr${}^{-1}$ [78].

As with DMSP-OLS, VIIRS-DNB data have been used to estimate the artificial component of night sky brightness at varying scales [79]. However, the DNB suffers a particularly acute shortcoming in sensing night lights attributable to solid-state lighting (SSL) sources on the ground, particularly those involving phosphor-converted “white” light-emitting diode (LED) light: it is completely insensitive to the white LED “blue peak” at ~0.45 μm, attributable to the blue LED source used to pump red and green emissions in the white LED package [80]. As the world continues to transition to SSL, the blindness of the VIIRS-DNB to a significant fraction of the light emissions of white LED is a serious disadvantage to quantifying the impact of this light source on the natural nighttime environment. There is a clear need for a dedicated orbital platform for night lights observing, and specific proposals addressing that need have been advanced with varying degrees of success [81,82,83].

Despite the shortcomings of facilities such as DMSP-OLS and VIIRS-DNB, both have been used to help identify proximate sources of skyglow in and adjacent to protected landscapes, particularly those seeking third-party certification for the quality of their night skies. An example illustrating the value of nighttime radiance imagery is shown in Figure 1. The four panels of the figure are centered near the Thompson Creek Mine, a large molybdenum mine in Idaho, U.S., and its adjacent tailing pond. VIIRS-DNB radiance data are overlaid on a daytime, visible-light satellite image of the area. A substantial decrease in light emissions from nighttime activities at the mine occurred between 2014 and 2015, when the mine ceased primary production of molybdenum ore and shifted operations to processing ore imported from South America. These images were used to help predict reductions to skyglow originating at the mine as part of the certification process for the nearby Central Idaho International Dark Sky Reserve and contribute to ongoing monitoring of external threats to the Reserve’s night sky quality.

Photographs of the nighttime side of the Earth obtained by astronauts aboard the International Space Station (ISS) are a third source of quantitative information about nights in and adjacent to protected places. The images are obtained with consumer-grade digital single reflex (DSLR) cameras and have yielded high-quality results [84]. Unlike data from dedicated orbital imagers, these photographs do not offer global coverage and sample individual locations irregularly in time. They are also typically not obtained toward nadir, requiring corrections due to the oblique viewing angles. Nevertheless, ISS images are especially useful in characterizing sensitive areas in urban contexts. Figure 2 shows as an example the city of Calgary, Canada, in a 2015 astronaut photo. Major waterways and a large municipal park are seen in silhouette against the surrounding city light. Such images help conservationists understand in remarkable detail the amount of light near these places, as well as the light color, which conveys information about the spectral power distribution (SPD) of the sources.

### 2.2. Single-Channel Radiometry

In addition to the above-mentioned shortcomings of satellite imagery of night lights, remote sensing techniques only yield estimates of night sky luminance from the ground that depend on model assumptions. Furthermore, these data do not address the experience of humans in places that are naturally dark at night, nor do they immediately reveal ALAN impacts to wildlife. For these purposes, we resort to direct measurements of night sky brightness obtained on the ground.

In the Twentieth Century, calibrated measurements of the brightness of the night sky were the exclusive province of astronomers, who had access to the equipment required to obtain and reduce luminance data with high photometric precision [85,86,87,88]. As early as the 1920s, some astronomers began experimenting with portable, purpose-built devices to measure sky brightnesses [89,90,91]. Extensive in situ measurements of zenith illuminance are now standard, enabled by the introduction of the Sky Quality Meter (SQM) in the early 2000s.

The SQM is a single-channel, networkable, temperature-compensated, frequency-counting photometer with a wide passband approximating the astronomical Johnson–Cousins *V* band [92], available in two versions of differing acceptance angles [93,94]. The devices have been shown to be photometrically stable over large ranges of temperature and rates of temperature change [95] and over intermittent use during periods of several years [96]. The response of SQMs to light is intercomparable among devices to within ±15% [97]. Furthermore, the device is available in a highly portable, battery-powered version designed for ease of use by otherwise untrained observers. Data-logging versions enable autonomous operation in the field, while networked versions provide real-time data from locations all over the world [98,99,100,101].

While the SQM and other devices of its kind excel at portability, enabling ground measurements from a diversity of locations, they remain single-channel instruments without meaningful angular resolution. By convention, sky quality is characterized by zenith measurements, which tend to overlook the contribution of horizon sources. Some approaches make use of multiple pointings of the SQM over a grid of points in altitude and azimuth to yield crude, interpolated maps of night sky brightness [102]. Furthermore, the wide spectral passband of the SQM and its spectral response in particular result in difficulty generalizing the conversion of radiance to luminance, making the interpretation of SQM data challenging [103]. Finally, the value of areal surveys performed with these devices is limited in the absence of a large number of monitoring stations. This limitation is partially overcome by schemes in which devices are attached to moving platforms, such as the automobile-based “Roadrunner” system [104].

### 2.3. Calibrated All-Sky Imagery

The lack of extended spatial resolution in single-channel detector measurements of night sky brightness is a clear limitation of the utility of those data. Increasingly, researchers recognize the value of detecting the brightness over the entire 2π steradians of sky through the use of calibrated, two-dimensional imaging techniques. A 2D imaging method allows for the extraction of a number of useful vector and scalar metrics from all-sky maps not possible with other methods [105]. 2D models of natural sources of light in the night sky can be subtracted from all-sky imagery, enabling the clear identification of distinctly anthropogenic sources [106].

Early efforts to deploy this technology involved purpose-built imaging devices with astronomical charge-coupled devices (CCDs) as their detectors [107,108,109]. More recently, experiments with astronomical CCDs [110] and lab-calibrated, commercially available DLSR camera and circular fisheye lens combinations [111] have yielded encouraging results. Novel uses for the method include applying differential photometry techniques to assess the impact of clouds on night skies over rural areas [112]; quantifying the effects on skyglow due to specific outdoor lighting interventions [113,114]; and exploring the potential for all-sky imaging from both stationary and moving platforms in marine, lacustrine, and littoral settings [55,115,116]. Recently published work extends this approach beyond sensing of only the upward (sky) hemisphere to include downward (ground) measurements [55]. It also considers the variable surface reflectivity in colder climates where the ground cover is dominated by snow and ice during parts of the year.

Current assessments of sites seeking third-party certification of their natural darkness resources and night sky quality combine both orbital and ground-based data sources, which helps identify specific sources of ALAN on local horizons and rank them in terms of threat significance. An example of an all-sky image associated with such a certification effort is shown in the top panel of Figure 3. The image was obtained from Tumacácori National Historical Park in Arizona, U.S., on 5 February 2018 during the park’s nomination process for IDA International Dark Sky Park status. It is matched with VIIRS-DNB annual cloud-free composite upward radiance data for southeast Arizona, U.S., in 2017 to illustrate the process by which individual “light domes” on the local horizon can be identified. The absence of artificial light indications on the VIIRS map toward the west-southwest of Tumacácori helps identify the diffuse light seen in that direction as a natural source: the zodiacal light, caused by scattering of sunlight from interplanetary dust in the plane of the Solar System.

## 3. Recent Developments

### 3.1. Drone-Based Aerial Imaging

As part of efforts to receive formal, third-party certification of sites for night sky quality, it is often necessary to create inventories of existing site lighting in order to identify problematic installations and create mitigation plans. This work has traditionally involved ground-based assessments of lighting augmented with sensing equipment such as DLSR cameras, illuminance meters, and handheld spectrometers. For large parks, in particular, this approach is labor- and time-intensive and runs the risk of failing to identify installations where their presence is not known in advance.

The proliferation of consumer-level unmanned aerial vehicles, also known as “drones,” is beginning to change the nature of the work of planning field data collection campaigns. Drone imagery is useful for locating installations, as seen in the pair of images in Figure 4. Geolocation with on-board GPS receivers helps pinpoint every instance of light in the scene, and color imagery provides an initial guess at lamp type and spectrum. The images indicate the number and extent of sources, which are then subjected to traditional ground-based validation. Future efforts in this direction may enable flying drones close enough to capture high-resolution images of luminaires in situ, day or night, and directly acquiring spectra and illuminance measurements.

### 3.2. Interpolated Single-Channel Detector Maps

Among the requirements for the accreditation of sites in the certification programs described in Section 1 is a characteristic night sky brightness, typically measured at the zenith, that does not exceed some threshold established by some semi-objective description of “dark” conditions [54,117]. Whether conducted with single-channel detectors like the SQM or all-sky imagery, sky quality surveys typically sample a small number of geographic locations in what are often large and sprawling candidate dark sky places. Large-scale areal maps of zenithal night sky brightness estimates can be produced from remote sensing measurements of upward nighttime radiance [1], but the accuracy of the estimates is model-dependent. While models can be improved by extensive ground validation, they fail to reproduce ground measurements faithfully for several reasons, including variable sky transparency due to turbid conditions, non-stable artificial sources, and the presence of natural sources of light in the night sky such as airglow.

The principal drawback of single-channel detectors is that they provide essentially no spatial information about the distribution of light across the night sky except in limited cases where measurements from many individual pointings of the device are interpolated to yield simple spatial maps of sky brightness (Figure 5). While the devices are often highly portable, interpreting the results from areal surveys is not necessarily straightforward [111,118]. In addition to the temporal sampling frequency of measurements to ensure robust statistics in the resulting maps, the reliability of the maps is controlled by the spatial sampling frequency. A rate on the order of one sample per square kilometer in a given region is necessary to constrain the uncertainty in zenithal night sky brightness measurements to within about ten percent [119].

### 3.3. Temporal Monitoring

Unattended night sky brightness monitoring stations offer the potential for gathering time-series observations of sky brightness with high temporal cadence. These datasets have value for following the evolution in sky brightness over protected areas. To date, these stations generally consist of single-channel detectors, such as networked and/or data-logging versions of the Sky Quality Meter. While they are shown to be reliable under field conditions, these types of devices are also limited in the kind of information they can provide. Calibrated all-sky imagery allows for identification of specific light sources on the horizon that are changing over time due to land use and development.

The two panels of Figure 6 show anthropogenic night sky luminance maps obtained from the same position in Theodore Roosevelt National Park in North Dakota, U.S., in 2010 and 2013. The mean all-sky luminance between the two epochs increased by more than a factor of five; this is due to vastly expanded activity during the interim associated with oil and natural gas extraction on the surrounding Bakken Shale formation. Maps like these help conservationists better understand the threat posed by ALAN emissions external (but adjacent) to protected landscapes. For now, the utility of imaging is limited by the degree to which human operators and data analysts are required in order to collect and process field data. In the future, these systems may achieve high reliability while operating autonomously in the field, adding significantly to their value to conservation practitioners.

## 4. Future Prospects

Even two decades after the establishment of designated programs by non-government organizations to recognize and certify the quality of night skies and nighttime darkness resources, the very notion of what a “dark sky” is remains unsettled from a scientific standpoint [117]; while appropriate instrumentation can quantify night sky brightness, it cannot properly account for the human aesthetic experience of natural night. However, various lines of research increasingly suggest that unsafe thresholds of exposure to artificial light at night in terms of intensity, duration, wavelength, and timing likely exist for humans, plants, and animals. In this sense, light-sensing technologies applied in the field could effectively serve as “dosimeters” for monitoring these exposure parameters.

Although they remain in wide use, it is clear that the utility of single-passband, single-point (zenith) measurements as a way of characterizing the overall night sky quality of a site seems to have reached its limit. The SPDs of commercially-available lighting products continue to evolve with time, due to changes in lamp technology, in situ aging of lighting equipment, and shifting consumer preferences; this has implications for making proper comparisons of, e.g., SQM measurements taken during different epochs [103]. To the extent that devices like the SQM remain the de facto standard for characterizing and monitoring the brightness of the night sky, especially over ecologically-sensitive locations, their utility for supporting interventions intended to promote conservation of dark night skies may be diminished. Recent pioneering work using hyperspectral imaging devices has yielded the ability to determine directly (and remotely) the SPDs of sources that contribute to anthropogenic skyglow on large spatial scales [120,121]. To the extent that these measurements are repeated in a given place over time, they can be used to determine the rate at which the source spectrum of skyglow is changing empirically.

The mean night sky brightness, averaged over the entire sky, is the most relevant parameter describing the impact of anthropogenic light at night on the nocturnal environment and yields a more reliable result than zenith luminance measurements alone, especially in places with low to moderate amounts of light pollution [105]. Such measurements are not easily obtained in the field with single-channel devices, much less those intended to operate autonomously. Furthermore, panchromatic imagery is preferable to luminance measurements in restricted passbands, presuming that the spectrum of the night sky will continue to change in the future [122]. In order to make future comparisons of like with like, the spectral response characteristics of detectors should be standardized to the greatest extent possible.

A need now clearly exists for something like a remotely-deployable, autonomous all-sky imaging system, at low cost and high ease of use, to more broadly portray and monitor nighttime conditions in protected places. Such a system could also provide spectrally-resolved sky radiance data across the entire visible sky, ideally with high angular resolution. Broader spectral coverage than what is currently possible with commercially available single-channel detectors would expand the applicability of sensing and monitoring to wavelength ranges beyond those most relevant to the human visual experience. Systems with enhanced ultraviolet and/or infrared responses are valuable in assessing ALAN effects on species that are acutely sensitive to these wavelengths, such as invertebrates and amphibians [123,124]. However, no such device with these characteristics is commercially available at present. Similarly, conservationists still lack an orbital platform that is sufficiently sensitive to short-wavelength visible light to monitor the ongoing global conversion to SSL; however, the conversion may well be complete by the time suitable data are available.

With improved data collection methods will come a need for more standardization of data collection, calibration, and reporting protocols, as well as better availability of software tools for reduction and analysis. There are developments in the latter, including both commercial [125] and open-source platforms [111,126,127]. It is further desirable to identify common laboratory calibration and field inter-comparison methods for these devices so as to ensure more reliable data obtained under real-world conditions. More uniform and reproducible data acquisition, reduction, and analysis procedures, as well as frequent calibration of devices against laboratory references will yield results that are more directly comparable across geographically disparate sites around the world.

Whether looking up or down, it is clear that imaging technologies will play an important role in future characterizations of light pollution and the impacts of ALAN on sensitive locations. The added capabilities of imagers, including increasing angular and spectral resolution, stand likely to revolutionize this field, especially as applied to time-domain studies. The further development of imaging techniques, equipment, and software is encouraged to provide the greatest benefit to both researchers and practitioners in measuring and monitoring the global influence of ALAN.

## Figures and Tables

**Figure 1 jimaging-05-00054-f001:**
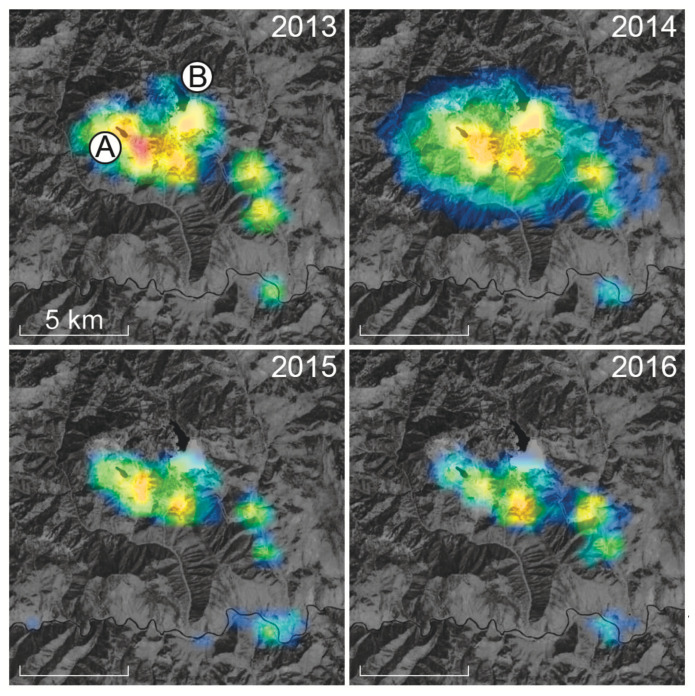
Evolution of the upward radiance from nighttime operations at the Thompson Creek Mine in Custer County, Idaho, U.S., between 2013 and 2017. In each of the four panels, color-coded Visible Infrared Imaging Radiometer Day-Night Band (VIIRS-DNB) radiance data from the National Oceanic and Atmospheric Administration/NASA Suomi National Polar-orbiting Partnership satellite are overlaid on a grayscale Landsat 8 Operational Land Imager context image obtained on 24 July 2016. The colors used to map the VIIRS-DNB radiance data range from 0.25 (darkest blue) to 40 (red) in units of nW cm${}^{-2}$ sr${}^{-1}$. The images are centered near 44${}^{\circ }$17′52.7″ N 114${}^{\circ }$31′21.7″ W and are oriented following the usual cartographic convention of north up, east right. A 5-km scale bar is shown in the lower left corner of each frame. The labels “A” and “B” in the upper-left panel indicate the open-pit molybdenum mine and its tailing pond, respectively.

**Figure 2 jimaging-05-00054-f002:**
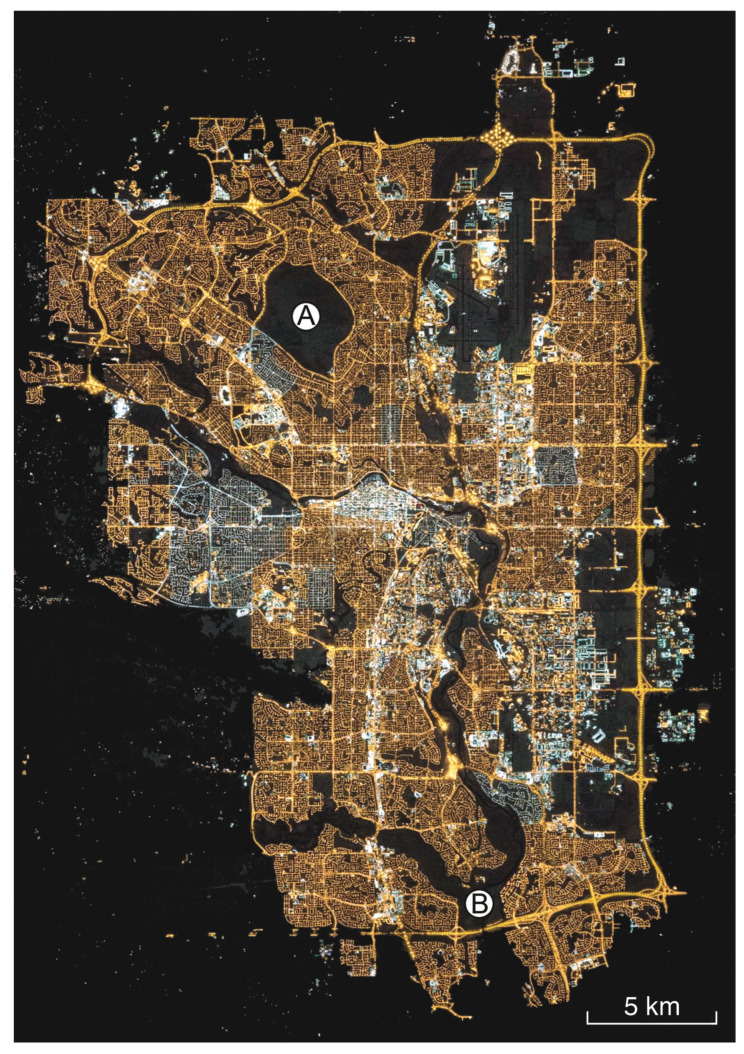
National Aeronautics and Space Administration photo ISS045-E-155026 of the city of Calgary, Canada, obtained by an astronaut aboard the International Space Station at 0707 UTC on 28 November 2015. The true-color image is oriented according to the usual cartographic convention with north up and east right, and a five-kilometer scale bar is provided in the lower right corner. The courses of major waterways, including the Bow River, are conspicuous as a series of sinuous silhouettes seen against the surrounding city lights. The image also shows two prominent protected areas within the city: Nose Hill Municipal Park (A) and Fish Creek Provincial Park (B).

**Figure 3 jimaging-05-00054-f003:**
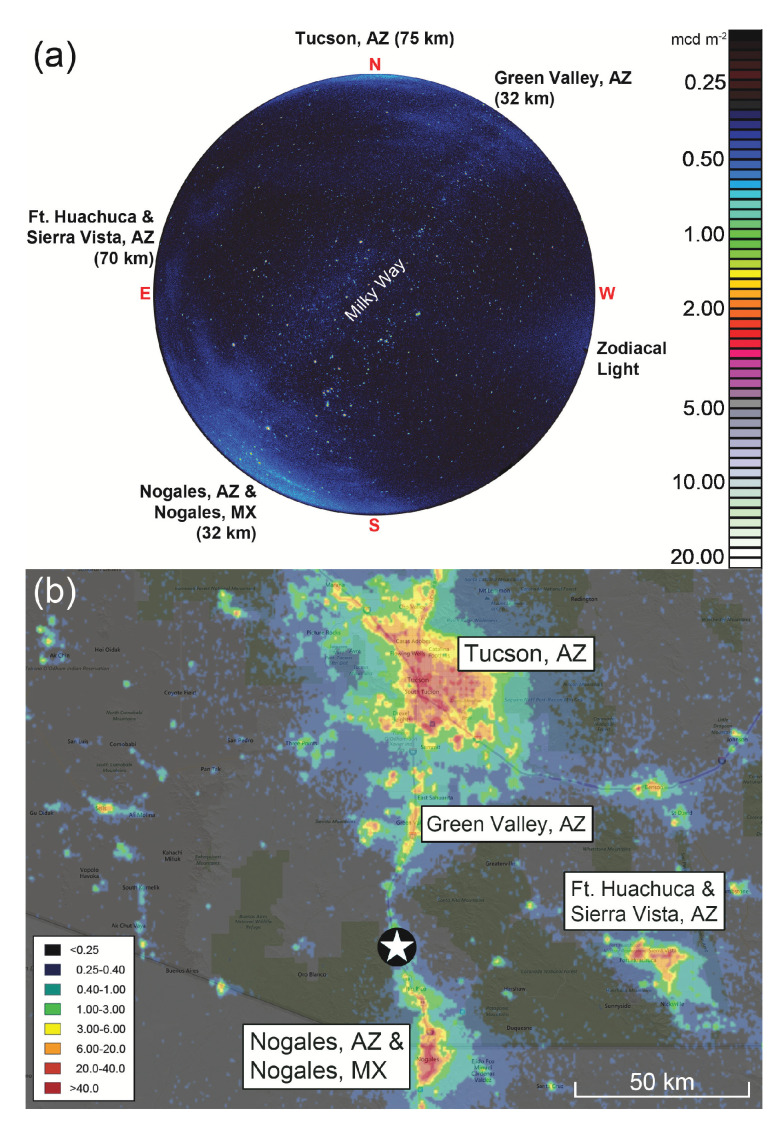
(**a**) An all-sky image of the night sky from Tumacácori National Historical Park in Arizona, U.S., obtained on the night of 5 February 2018. The view is centered on the zenith, and the cardinal points on the horizon are indicated in red letters. The image orientation follows the astronomical convention for all-sky imagery (north at top and east left). The color bar gives sky luminances in units of mcd m${}^{-2}$. Sources of natural and artificial light are labeled; anthropogenic sources and their radial distances from the site are listed. (**b**) VIIRS-DNB annual cloud-free composite upward radiance data (false colors) for southeast Arizona, U.S., in 2017 overlaid on a Google Map base. Radiances are given in units of nW cm${}^{-2}$ sr${}^{-1}$ according to the color key at the lower left, and a 50-kilometer scale bar is shown at the lower right. North is up, and east is right, according to the usual cartographic convention. The location of Tumacácori National Historical Park is indicated by the five-pointed star at the bottom-center. Other locations referenced in (a) are labeled. Background map copyright 2019 Google, INEGI, used with permission.

**Figure 4 jimaging-05-00054-f004:**
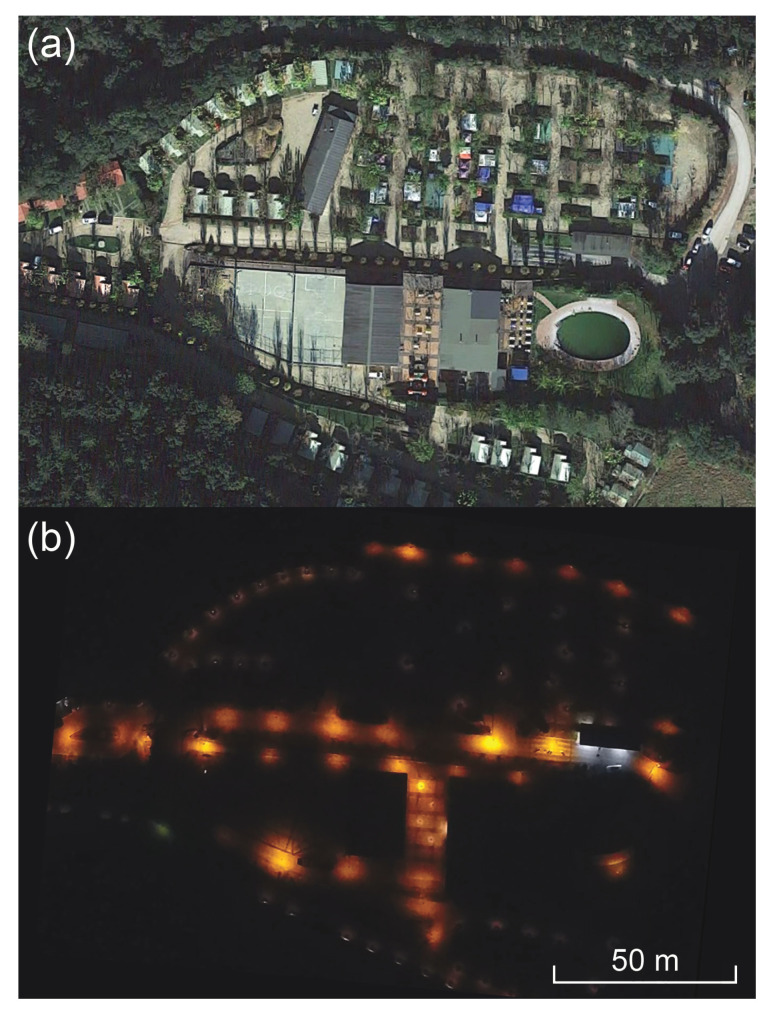
An application of drone-based aerial imagery to the identification of artificial light sources on the ground. (**a**) A daytime satellite image of Bassegoda Park, a camping facility in Girona, Catalonia. Image copyright 2019 Google, INEGI, used with permission. (**b**) A color digital image of the site at night. The image was obtained by a drone flying at an altitude of approximately 100 m above the site. The locations of individual artificial light sources are identified by “pools” of light beneath each, indicating reflection from the ground. The scene is dominated by “warm,” low-pressure sodium lighting, except for two tube fluorescent luminaires near the park entrance station (white sources at right). North is up and east right in both images, and the 50-m scale bar in the lower right corner is common to both.

**Figure 5 jimaging-05-00054-f005:**
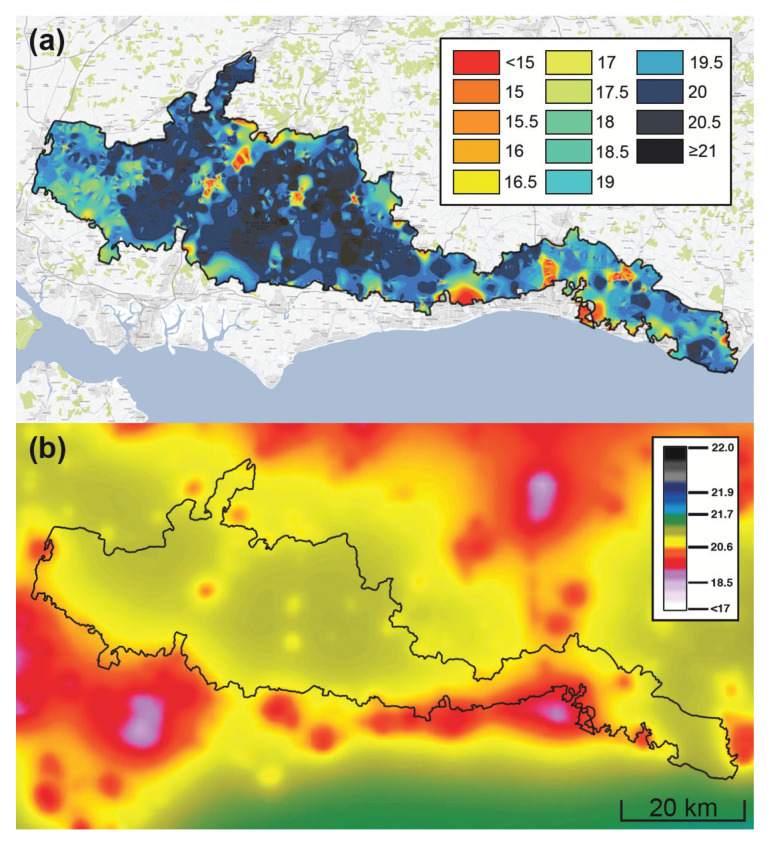
Two methods of estimating the spatial distribution of zenithal night sky brightness over South Downs National Park, England. (**a**) Interpolated map of over 20,000 individual Sky Quality Meter measurements obtained in 2014–2015 during the park’s bid for International Dark-Sky Association (IDA) International Dark Sky Reserve status. (**b**) Map of implied visual-band zenith luminance derived from remote sensing measurements of upward radiance described in [1] during 2015. In both panels, the park boundary is indicated by the solid black line. The false colors in both maps indicate the sky luminance in units of magnitudes per square arcsecond; note that the colors in both maps correspond to different ranges of luminance. The 20-km scale bar at the lower right is common to both maps, and both are oriented according to the usual cartographic convention (north up, east right).

**Figure 6 jimaging-05-00054-f006:**
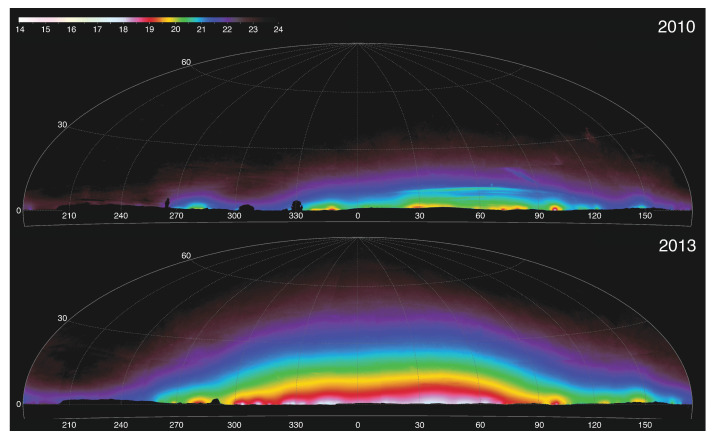
Two views of the modeled anthropogenic component of night sky brightness as seen from Oxbow Overlook in Theodore Roosevelt National Park, U.S., on 1 October 2010 (top) and 9 May 2013 (bottom). These all-sky maps, centered on north (0${}^{\circ }$ azimuth), are rendered in the Hammer–Aitoff projection with lines of constant altitude and azimuth shown. Calibrated night sky luminance data were obtained using the method described in [109], and a model of the natural sources of light in the night sky according to [106] was subtracted to yield the angular distribution of anthropogenic light. The false colors indicate luminance in units of visual magnitudes per square arcsecond, indicated by the color bar at the upper left. The data were obtained and calibrated by the U.S. National Park Service Natural Sounds and Night Skies Division.

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
