# Peer review of "Methods for Assessment and Monitoring of Light Pollution around Ecologically Sensitive Sites"

_2313-433X, 2019, doi:10.3390/jimaging5050054_

Round 1

Reviewer 1 Report

Article: Imaging and Photometric Assessment and Monitoring of Light Pollution Impacts to Ecologically Sensitive Sites

By John C. Barentine

In this article, the author makes an overall review of the different imaging methods to monitor sky brightness and light pollution. I think the author makes a good compilation of methods which can be useful for non-expert readers. I miss case studies in which imaging assessment of light pollution has served to report or mitigate its effect on ecological systems, species or processes.  

Minor changes:

Title: Is the title correct? As a non-native English speaker the two “and” sound very strange to me.

Lines 44-49: Please introduce here also the wildlife ecotourism. Nocturnal wildlife watching should be done in the most natural conditions. See for example the penguin watching experience at Australia and New Zealand, which is partially covered in this paper:

Rodríguez, A, Holmberg, R, Dann, P, Chiaradia, A. Penguin colony attendance under artificial lights for ecotourism. J Exp Zool Part A. 2018; 329: 457– 464. https://doi.org/10.1002/jez.2155

Line 59: Delete “in order”

Line 60: Please re-word. “…to adjacent protected areas” but also to urban areas and unprotected areas.

Lines 245-249: It seems that figure 6 is not very relevant. Some reserves show a high variation in the measures and without other methodological details, it is not very informative. It is partially recognized by the author. I would like to see additional data of other imaging assessments (e.g. VIIRS, drones, or ISS imagery) for a comparison.

Line 264: “…likely exist for both humans and animals” I would say biodiversity to include also plants.

Figure 6: What do whiskers indicate?

Author Response

Author responses to jimaging-492495 Reviewer 1 report

7 May 2019

Instances where I agree with the reviewer and have made the change appear in green.

Instances where I disagree with the reviewer, and the reasons for the disagreement, appear in red.

Reviewer 1

In this article, the author makes an overall review of the different imaging methods to monitor sky brightness and light pollution. I think the author makes a good compilation of methods which can be useful for non-expert readers. I miss case studies in which imaging assessment of light pollution has served to report or mitigate its effect on ecological systems, species or processes.  

Minor changes:

Title: Is the title correct? As a non-native English speaker the two “and” sound very strange to me.

Although I feel that the original title reads all right, at least from the perspective of a native English speaker, I certainly am amenable to other options. After considering this comment, and the similar statement by Reviewer 3, I have changed the title to “Methods for Assessment and Monitoring of Light Pollution Around Ecologically Sensitive Sites”. I think this keeps the sense of the original, while, as Reviewer 3 points out, does not imply new/original results involving ecological impacts of light at night.

Lines 44-49: Please introduce here also the wildlife ecotourism. Nocturnal wildlife watching should be done in the most natural conditions. See for example the penguin watching experience at Australia and New Zealand, which is partially covered in this paper:

Rodríguez, A, Holmberg, R, Dann, P, Chiaradia, A. Penguin colony attendance under artificial lights for ecotourism. J Exp Zool Part A. 2018; 329: 457– 464. https://doi.org/10.1002/jez.2155

I am happy to make this addition. Other than the reference to Rodríguez et al. (2018), there is little in the tourism research literature that specifically addresses practices related to nighttime wildlife watching.

The paragraph previously read:

Parks, nature reserves and similar protected areas around the world now find themselves at the forefront of the effort to preserve what remains of the planet's natural nighttime darkness. In many instances, they are equally motivated by the economic development potential of sustainable ‘astrotourism' catering to visitors who wish to view dark night skies. However, best practices for managing this resource are still nascent, and the literature on the subject is scant and tends to focus on tourism rather than systematic conservation practices.

It now reads:

Parks, nature reserves and similar protected areas around the world now find themselves at the forefront of the effort to preserve what remains of the planet's natural nighttime darkness. In many instances, they are equally motivated by the economic development potential of sustainable forms of tourism. These tourism modalities include ‘astrotourism' catering to visitors who wish to view dark night skies, as well as wildlife ecotourism, in which visitors seek to view nocturnal species in natural nighttime conditions. However, best practices for managing this resource are still nascent, and the literature on the subject is scant and tends to focus on tourism rather than systematic conservation practices. Where wildlife is concerned, there is some evidence that lighting provided for the convenience of ecotourists is potentially detrimental to the species they come to view; however, other work suggests that actively managing nighttime visitor interactions with wildlife can minimize potential impacts while increasing visitor satisfaction.

In addition to the reference suggested by the reviewer, I added a reference to Wolf and Croft (Tourism Management Perspectives, Vol. 4, October 2012, Pages 164-175) in which a study at an Australian rangeland tourist site is described that considered factors including outdoor lighting.

Line 59: Delete “in order”

This change is made in the revision.

Line 60: Please re-word. “…to adjacent protected areas” but also to urban areas and unprotected areas.

This change is made in the revision. The sentence now reads “The same is true of cities seeking accreditation as IDA International Dark Sky Communities: it is important to know the magnitude of light pollution in and near cities to assess the efficacy of public policies and lighting interventions intended to reduce cities' impacts not only to urban areas and places lacking environmental protections, but also to adjacent protected areas.”

Lines 245-249: It seems that figure 6 is not very relevant. Some reserves show a high variation in the measures and without other methodological details, it is not very informative. It is partially recognized by the author. I would like to see additional data of other imaging assessments (e.g. VIIRS, drones, or ISS imagery) for a comparison.

While I disagree with the reviewer about the relevance of the figure, in recent days I have come to re-evaluate its utility in this particular paper in light of a manuscript currently under preparation. In that paper, which contains an analysis of remote sensing data from the VIIRS Day-Night Band aboard the Suomi NPP satellite, the authors find that measurements of International Dark Sky Places at least since the launch of the satellite in 2011 are insufficient to address their hypotheses about how lighting policies in the Places affect light emissions. Their result is rather like the figure that the reviewer called “not very relevant”.

I have concluded that there simply does not yet exist a set of data, either involving sky brightness measurements from the ground or upward radiance detected from orbit, that can tell us anything about lighting trends in these protected areas. For that reason, I will withdraw the figure, while leaving in place the former Figure 7, now relabeled as Figure 6. I do not see that any essentially point of the paper is no longer adequately made in the absence of this figure.

Accordingly, I have changed the text in the subsection 3.3 (“Temporal monitoring”). Up to the discussion of the (newly numbered) Figure 6, the text now reads:

Unattended night sky brightness monitoring stations offer the potential for gathering time-series observations of sky brightness with high temporal cadence. These data sets have value for following the evolution in sky brightness over protected areas. To date, these stations generally consist of single-channel detectors, such as networked and/or data-logging versions of the Sky Quality Meter. While they are shown to be reliable under field conditions, these types of devices are also limited in the kind of information they can provide. Calibrated all-sky imagery allows for identification of specific light sources on the horizon that are changing over time due to land use and development.

I have also added two sentences to the end of the following paragraph:

For now, the utility of imaging is limited by the degree to which human operators and data analysts are required in order to collect and process field data. In the future these systems may achieve high reliability while operating autonomously in the field, adding significantly to their value to conservation practitioners. 

I believe this adequately addresses the concern of the reviewer.

Line 264: “…likely exist for both humans and animals” I would say biodiversity to include also plants.

This change is made in the revision. The sentence now reads: “However, various lines of research increasingly suggest that unsafe thresholds of exposure to artificial light at night in terms of intensity, duration, wavelength and timing likely exist for humans, plants and animals.” In the sentence that follows, we have exchanged the word “imaging” with “light-sensing”, the latter of which is more factually accurate (as the techniques could include non-imaging photometry).

Figure 6: What do whiskers indicate?

Now moot, given the deletion of the figure mentioned previously.

Reviewer 2 Report

Very nicely done!

Author Response

Author responses to jimaging-492495 Reviewer 2 report

7 May 2019

Instances where I agree with the reviewer and have made the change appear in green.

Instances where I disagree with the reviewer, and the reasons for the disagreement, appear in red.

Reviewer 2

No request for changes were made by this reviewer.

Reviewer 3 Report

I have enjoyed reading this Review article on "Imaging and Photometric Assessment and Monitoring of Light Pollution Impacts to Ecologically Sensitive Sites". The summary and the description of the state of the art may be highly useful for new researchers entering this field, as well as for fostering a necessary debate on the short- and mid-term priorities faced by the Light Pollution research community regarding site monitoring techniques. These are in my opinion the major merits of this work, and, even if some of the views of the Author could be debatable (or precisely because of that), I have no objection against publishing it basically in its present form.

My main comments, that the Author may want to consider for improving this paper but I do not formulate as a pre-requisite for acceptance, are as follows:

- I would suggest removing the word "Impacts" from the title, just to avoid potential readers expecting finding in it some specific new developments regarding ecological impacts on individuals, populations or ecosystems. Most of the paper deals with monitoring of light pollution, rather than with its impacts in strict sense.

- I feel that the last section ("Future Prospects") still has room for a bit further elaboration. I agree with the points raised by the Author, but I believe readers would appreciate a slightly more extended description of the challenges that the LP research community should address in this field.

Regarding references:

- Given the review character of this work, I missed several references by M. Kocifaj that could be even more interesting than the one already included [62]. These are:

Kocifaj M. (2007). Light-pollution model for cloudy and cloudless night skies with ground-based light sources, Applied Optics 46, 3013-3022
Kocifaj M. (2016). A review of the theoretical and numerical approaches to modeling skyglow: Iterative approach to RTE, MSOS, and two-stream approximation, Journal of Quantitative Spectroscopy & Radiative Transfer 181, 2–10
Kocifaj M. (2018). Multiple scattering contribution to the diffuse light of a night sky: A model which embraces all orders of scattering, Journal of Quantitative Spectroscopy and Radiative Transfer 206, 260-272 https://doi.org/10.1016/j.jqsrt.2017.11.020

- A http or doi for refs [99] and [101], if available, would be useful for readers.

Author Response

Author responses to jimaging-492495 Reviewer 3 report

7 May 2019

Instances where I agree with the reviewer and have made the change appear in green.

Instances where I disagree with the reviewer, and the reasons for the disagreement, appear in red.

Reviewer 3

I have enjoyed reading this Review article on "Imaging and Photometric Assessment and Monitoring of Light Pollution Impacts to Ecologically Sensitive Sites". The summary and the description of the state of the art may be highly useful for new researchers entering this field, as well as for fostering a necessary debate on the short- and mid-term priorities faced by the Light Pollution research community regarding site monitoring techniques. These are in my opinion the major merits of this work, and, even if some of the views of the Author could be debatable (or precisely because of that), I have no objection against publishing it basically in its present form.

My main comments, that the Author may want to consider for improving this paper but I do not formulate as a pre-requisite for acceptance, are as follows:

- I would suggest removing the word "Impacts" from the title, just to avoid potential readers expecting finding in it some specific new developments regarding ecological impacts on individuals, populations or ecosystems. Most of the paper deals with monitoring of light pollution, rather than with its impacts in strict sense.

See previous comments in the response to Reviewer 1 about this.

- I feel that the last section ("Future Prospects") still has room for a bit further elaboration. I agree with the points raised by the Author, but I believe readers would appreciate a slightly more extended description of the challenges that the LP research community should address in this field.

This request is reasonable. I have increased the content of this section by about 50% in terms of words, and added nine additional references. The added content adds additional discussion about the problem of evolving spectral power distributions of skyglow sources over time, and ways that the changes could be monitored. In addition, I make a plea for the standardization of the practices around the acquisition, calibration, and interpretation of data to increase the comparability of measurements taken in different places around the world and at different times. I hope these additions are of the variety that the reviewer is looking for in a revised manuscript.

Regarding references:

- Given the review character of this work, I missed several references by M. Kocifaj that could be even more interesting than the one already included [62]. These are:

Kocifaj M. (2007). Light-pollution model for cloudy and cloudless night skies with ground-based light sources, Applied Optics 46, 3013-3022

Kocifaj M. (2016). A review of the theoretical and numerical approaches to modeling skyglow: Iterative approach to RTE, MSOS, and two-stream approximation, Journal of Quantitative Spectroscopy & Radiative Transfer 181, 2–10

Kocifaj M. (2018). Multiple scattering contribution to the diffuse light of a night sky: A model which embraces all orders of scattering, Journal of Quantitative Spectroscopy and Radiative Transfer 206, 260-272 https://doi.org/10.1016/j.jqsrt.2017.11.020

- A http or doi for refs [99] and [101], if available, would be useful for readers.

Neither of these references has a doi. In lieu of this, I have added to the reference in each case a URL at which the resource can be found on the web. This has the disadvantage of the potential for ‘link rot,’ but I think it’s the best that I can do for readers at this point in time.

I have also reviewed the BibTeX file accompanying the submission to correct instances where extraneous information was provided in the references. In some cases, I was able to add DOIs to entries where they did not exist in the original submission.

Round 2

Reviewer 1 Report

I think that the author has properly addressed all the reviewer questions. For that reason, I vote for the acceptation of this manuscript in its present form. I only have a minor suggestion. Please, delete all the "in order to" by "to" (three times in the ms). "in order" does not provide any information. Avoiding its use is recommended In scientific literature (see for example Valiela, I. 2009 Doing science: design, analysis, and communication of scientific research. 2nd edition. Oxford University Press).